# Deep model-based optoacoustic image reconstruction (DeepMB)

**Christoph Dehner**[*,1,2]                                    CHRISTOPH.DEHNER@ITHERA-MEDICAL.COM

**Guillaume Zahnd**[*,1,2]                                    GUILLAUME.ZAHND@ITHERA-MEDICAL.COM

**Vasilis Ntziachristos**[†,1,3,4]                            BIOIMAGING.TRANSLATUM@TUM.DE

**Dominik Jüstel**[†,1,3,5]                                   DOMINIK.JUSTEL@HELMHOLTZ-MUNICH.DE

[1] *Institute of Biological and Medical Imaging, Helmholtz Zentrum München, Neuherberg, Germany*

[2] *iThera Medical GmbH, Munich, Germany*

[3] *Chair of Biological Imaging at the Central Institute for Translational Cancer Research (Transla-TUM), School of Medicine, Technical University of Munich, Germany*

[4] *Munich Institute of Robotics and Machine Intelligence, Technical University of Munich, Germany*

[5] *Institute of Computational Biology, Helmholtz Zentrum München, Neuherberg, Germany*

## Abstract

Multispectral optoacoustic tomography requires image feedback in real-time to locate and identify relevant tissue structures during clinical interventions. Backprojection methods are commonly used for optoacoustic image reconstruction in real-time but only afford imprecise images due to oversimplified modelling assumptions. Herein, we present a deep learning framework, termed DeepMB, that infers optoacoustic images with state-of-the-art quality in 31 ms per image.

**Keywords:** Multispectral Optoacoustic Tomography (MSOT), Model-based reconstruction, Inverse problems, Real-time imaging, Synthesized training data

## 1. Introduction

Multispectral optoacoustic tomography (MSOT) can non-invasively detect optical contrast in living tissue with high spatial resolution and centimeter-scale penetration depth. Similar to ultrasound imaging, clinical use of MSOT requires image feedback in real-time to locate and identify relevant tissue structures. Backprojection methods (Xu and Wang, 2005) can reconstruct optoacoustic images in real-time but only deliver imprecise images due to oversimplified modelling assumptions. On the other hand, iterative model-based reconstruction (Chowdhury et al., 2020, 2021) delivers state-of-the-art optoacoustic image quality. However, the required computational effort and the iterative approach of the algorithm prevent it from being used for real-time imaging. Deep learning enables faster image reconstruction using deep neural network models that support efficient and GPU-accelerated inference, however the lack of experimental ground truth training data can lead to reduced image quality for in vivo data (Kim et al., 2020; Hauptmann and Cox, 2020; Gröhl et al., 2021).

---

[*] Contributed equally

[†] Shared corresponding authorship

Herein, we show that learning a well-posed reconstruction operator facilitates accurate generalization from synthesized training data to experimental test data. We present a deep learning framework, termed DeepMB, that infers optoacoustic images with quality nearly-indistinguishable from state-of-the-art model-based reconstructions at speeds enabling live imaging (31 ms per image). DeepMB facilitates accurate model-based reconstruction for arbitrary experimental input data through training on optoacoustic signals synthesized from real-world images, while using as ground truth for the first time the optoacoustic images generated by model-based reconstruction of the corresponding signals.

## 2. Methods

Figure 1 illustrates the training and evaluation process of DeepMB. Input sinograms for network training were obtained by utilizing general-feature images (Everingham et al., 2009) as initial pressure distributions and simulating thereof the signals recorded by the acoustic transducers with an accurate physical model of the scanner (Fig. 1a). In vivo sinograms for evaluating the performance of the trained network were acquired by scanning six participants at up to eight anatomical locations each (Fig. 1b). Ground truth images for both the synthetic training sinograms and the in vivo test sinograms were generated using model-based reconstruction (Fig. 1c). The deep neural network used for DeepMB consists of a delay operation, followed by trainable (U-net-like) convolutional layers (Fig. 1d). The network was trained end-to-end on synthesized input sinograms and corresponding model-based reference images for 300 epochs using stochastic gradient descent.

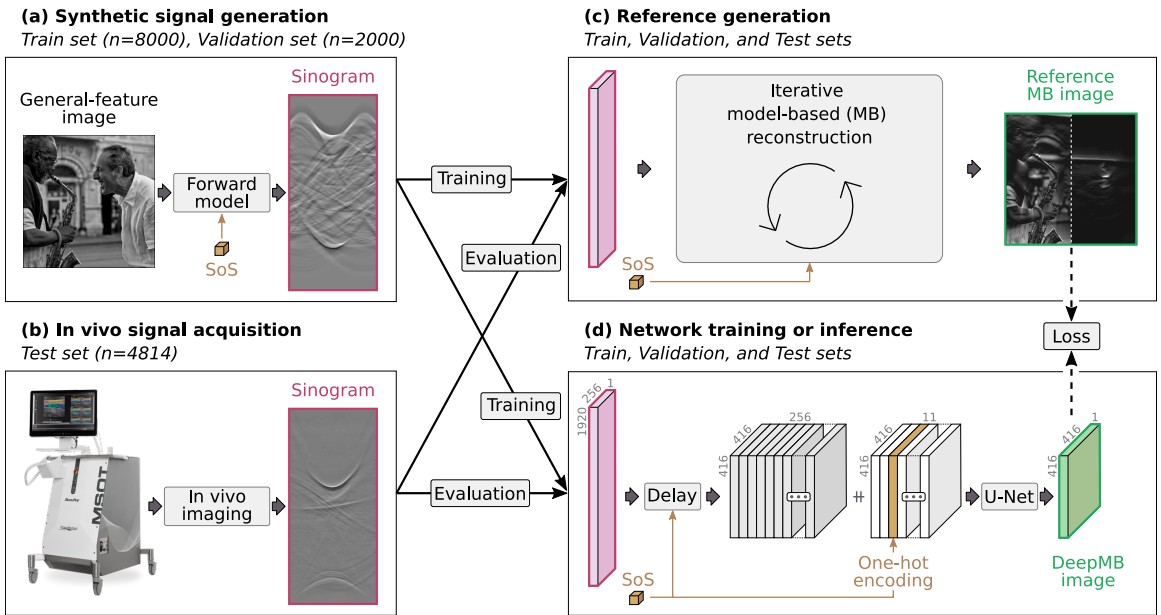

Figure 1: Training and evaluation process of DeepMB.

## 3. Results

DeepMB infers optoacoustic images in 31 ms per sample using a recent graphics processing unit (NVIDIA GeForce RTX 3090). The performance of DeepMB was evaluated using 4814 in vivo sinograms that were acquired with a modern clinical optoacoustic scanner (MSOT Acuity Echo, iThera Medical GmbH, Munich, Germany). Figure 2 shows the optoacoustic images from model-based, DeepMB, and backprojection reconstruction for a scan of a human carotid. DeepMB images are systematically nearly-indistinguishable from model-based references. In contrast, backprojection images suffer from reduced spatial resolution and physically-nonsensical negative initial pressure values.

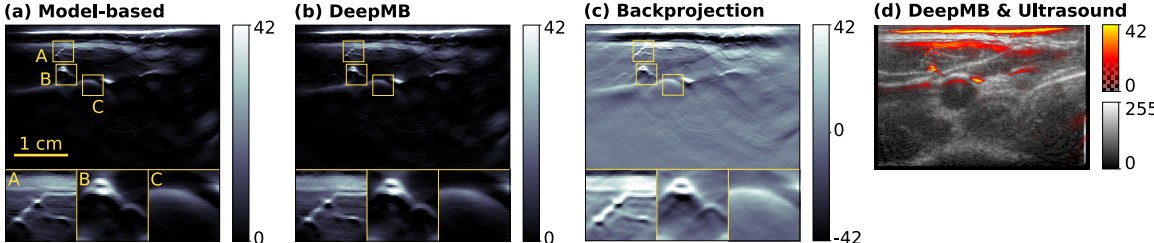

Figure 2: Optoacoustic images from model-based, DeepMB and backprojection reconstruction for a scan of human carotid at 800 nm.

Table 1 summarizes a quantitative comparison of model-based, DeepMB, and backprojection images. The obtained metrics confirm that the image quality of DeepMB is comparable to model-based reconstruction and superior to backprojection reconstruction.

Table 1: Quantitative evaluation of the image quality for all 4814 in vivo sinograms from the test dataset (mean value, [25$^{\text{th}}$ and 75$^{\text{th}}$ percentiles]).

|  |  | Reference method Model-based | Our method DeepMB | Traditional method Backprojection |
|---|---|---|---|---|
| Data residual norm | ($\downarrow$) | 0.139 [0.068, 0.180] | 0.156 [0.092, 0.189] | 0.369 [0.294, 0.428] |
| Mean square error | ($\downarrow$) | n/a | 9.45 [0.56, 2.41] | 84.98 [24.97, 85.20] |
| Structural similarity | ($\uparrow$) | n/a | 0.98 [0.98, 0.99] | 0.73 [0.68, 0.79] |

## 4. Conclusion

DeepMB can enable state-of-the-art MSOT imaging in clinical applications that require real-time image feedback. The source code of DeepMB is available on GitHub[1], and further details are described in our arXiv preprint (Dehner et al., 2022). We are currently working on integrating DeepMB into the hardware of a next-generation MSOT scanner.

---

1. https://github.com/juestellab/deepmb

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
