# OpenReview forum: "Deep model-based optoacoustic image reconstruction (DeepMB)"
_MIDL.io/2023/Short_Paper_Track — MIDL 2023 Short paper track Poster_

### Official Review · Reviewer_Qp8h · 2023-04-12
**Interesting solution to a real-world medical imaging problem**

**Rating:** 8
**Confidence:** 4

**Review:**

This short paper describes a deep-learning approach for optoacoustic image reconstruction in real time. Authors train the method using abundant natural images and validate using in-vivo images. As a reference, a state-of-the-art but slow model-based reconstruction method is used. Because there is no other ground truth, the deep learning approach is trained to mimic the model-based approach. The method is shown to outperform FBP. This is a well-written paper on a problem that may not be known to all MIDL attendees. As such, it would be of added value to the conference.

Strengths
- The paper addresses a real and challenging problem, i.e., real-time image reconstruction
- The use of non-medical images for training is interesting and does not seem to hamper performance in in-vivo imaging
- The short paper is well-written and easy to follow
- Authors provide code

Weaknesses
- It remains unclear why exactly the proposed method is so fast; some more technical details would have been welcome
- In essence, it appears that the reconstruction method is simply a U-Net applied to delayed reconstruction, which in itself is not very novel

---

### Official Review · Reviewer_QDom · 2023-04-25
**Interesting approach and reasonable evaluation, not clear how this approach advances the state of the art in an active field.**

**Rating:** 7
**Confidence:** 4

**Review:**

This paper presents an approach to optoacoustic image reconstruction that uses general image features to drive the reconstruction via a U-Net. The model is trained against model-based reconstructions. The authors show that the reconstructions from real signals match very closely model-based reconstructions.

The approach and the evaluation are reasonable. The paper is clearly written. It was unclear to me how the demonstrated reconstructions compare to the state of the art methods in this active research field. The authors cite a review paper from 2021, which describes many deep learning approaches to optoacoustic reconstruction. IS this the first time someone trains against a model-based reconstruction rather than ground truth images in the simulated data? How does the reconstruction compare to the ground truth in segmentations? Are model-based reconstructions so good that we should accept them as ground truth? I realize the space is tight, but just a sentence or two about the main advance here relative to the state of the art would significantly strengthen the paper.